# Ciprofloxacin Concentrations 1/1000th the MIC Can Select for Antimicrobial Resistance in *N. gonorrhoeae*—Important Implications for Maximum Residue Limits in Food

**DOI:** 10.3390/antibiotics11101430

**Published:** 2022-10-18

**Authors:** Natalia González, Saïd Abdellati, Irith De Baetselier, Jolein Gyonne Elise Laumen, Christophe Van Dijck, Tessa de Block, Sheeba Santhini Manoharan-Basil, Chris Kenyon

**Affiliations:** 1STI Unit, Department of Clinical Sciences, Institute of Tropical Medicine, 2000 Antwerp, Belgium; 2Clinical Reference Laboratory, Department of Clinical Sciences, Institute of Tropical Medicine, 2000 Antwerp, Belgium; 3Laboratory of Medical Microbiology, University of Antwerp, 2610 Wilrijk, Belgium; 4Division of Infectious Diseases and HIV Medicine, University of Cape Town, Cape Town 7700, South Africa

**Keywords:** *N. gonorrhoeae*, antimicrobial consumption, AMR, resistance, quinolone, one-health

## Abstract

Background: Concentrations of fluoroquinolones up to 200-fold lower than the minimal inhibitory concentration (MIC) have been shown to be able to select for antimicrobial resistance in *E. coli* and *Salmonella* spp. (the minimum selection concentration—MSC). We hypothesized that the low concentrations of quinolones found in meat may play a role in the genesis of quinolone resistance in *Neisseria gonorrhoeae*. We aimed to (i) establish the ciprofloxacin MSC for *N. gonorrhoeae* and (ii) assess if, at the ecological level, the prevalence of gonococcal ciprofloxacin resistance is associated with the concentration of quinolones used in food animal production, which is an important determinant of long-term low-dose exposure to ciprofloxacin in humans. Methods: (i) To assess if subinhibitory ciprofloxacin concentrations could select for de novo generated resistant mutants, a susceptible WHO-P *N. gonorrhoeae* isolate was serially passaged at 1, 1:10, 1:100 and 1:1000 of the ciprofloxacin MIC of WHO-P (0.004 mg/L) on GC agar plates. (ii) Spearman’s correlation was used to assess the association between the prevalence of ciprofloxacin resistance in *N. gonorrhoeae* and quinolone use for animals and quinolone consumption by humans. Results: Ciprofloxacin concentrations as low as 0.004 µg/L (1/1000 of the MIC of WHO-P) were able to select for ciprofloxacin resistance. The prevalence of ciprofloxacin resistance in *N. gonorrhoeae* was positively associated with quinolone use for food animals (ρ = 0.47; *p* = 0.004; N = 34). Conclusion: Further individual level research is required to assess if low doses of ciprofloxacin from ingested foodstuffs are able to select for ciprofloxacin resistance in bacteria colonizing humans and other species.

## 1. Introduction

Gonococcal resistance to fluoroquinolones has spread around the world, but the prevalence of resistance varies considerably between countries and regions. Certain countries in East Asia are amongst those most heavily affected. The prevalence of ciprofloxacin resistance in China, for example, increased from 10% to 95% between 1996 and 2003 [1]. In comparison, the country-level median prevalence of ciprofloxacin resistance in 2009 was 24% in the Americas and 6% in Africa [2]. Whilst differences in the intensity of exposure to fluoroquinolones likely play a key role in explaining these differences in resistance, it is unclear what pathways are most important. Since *Neisseria gonorrhoeae* is asymptomatic the majority of the time it circulates in a population, quinolones used for other indications can also select for antimicrobial resistance (AMR), i.e., bystander selection [3]. *N. gonorrhoeae* can also acquire resistance conferring genes from commensal *Neisseria* [4]. Quinolone used for any indication could select for ciprofloxacin resistance in commensals and, subsequently, in *N. gonorrhoeae,* i.e., indirect selection [5,6]. Studies have provided support for each of these pathways in the genesis of gonococcal AMR [4,5,6]. In particular, a number of studies have established that a country-level consumption of quinolones is a predictor of ciprofloxacin resistance [2,3]. Certain Asian countries such as China, Malaysia and the Philippines have been noted to be outliers in these analyses [2]. In 2009, for example, these three countries had close to 100% resistance to ciprofloxacin. However, all consumed less than half the volume of quinolones consumed by the United States, where less than 10% of gonococcal isolates were resistant to ciprofloxacin [2]. 

There is also an increasing body of evidence suggesting that quinolone resistance in *N. gonorrhoeae* in Asia and elsewhere is part of a syndemic of resistance affecting a range of Gram-negatives, including *Escherichia coli* and *Pseudomonas* spp. [7]. Further evidence for the syndemic perspective comes from studies that show how the prevalence of quinolone resistance is considerably higher (up to 16-fold higher) in several Gram-negatives in Asia compared to in Europe and the Americas [8,9,10,11]. Between 1998 and 2009, for example, the prevalence of ciprofloxacin resistance in *Shigella* increased from 0% to 29% in Asia compared to 0% to 0.6% in Europe–America [8]. This syndemic of quinolone resistance included *Neisseria meningitidis* and various species of commensal *Neisseria* [12]. Recent studies from Shanghai, for example, found the prevalence of ciprofloxacin resistance to be 99% in *N. gonorrhoeae*, 100% in commensal *Neisseria* and 66% in *N. meningitidis* [6,13,14]. These findings suggest that unconsidered types of quinolone exposure may play a role in the genesis and spread of quinolone resistance. One such possibility is the use of quinolones in animal husbandry. 

Quinolone use in animals has been linked to AMR in a number of Gram-negative pathogens circulating in humans [15,16]. This use of quinolones could induce resistance in *N. gonorrhoeae* directly or indirectly. Direct selection on *N. gonorrhoeae* could occur via human ingestion of quinolone residues in meats or water/soil contaminated by animal manure [17]. Quinolones have been found to show very low biodegradability in the environment [17,18]. Selection could also occur indirectly where quinolones select for resistance in commensal *Neisseria,* which would then be transformed into *N. gonorrhoeae*. Commensal *Neisseria* have been found in the resident microbiomes in a range of food animals, including chickens, cows, sheep and goats [19,20,21,22]. The selection of quinolone resistance in commensal *Neisseria* could thus occur in animals or humans. Of note, the indirect pathway has been shown to be important in the genesis and spread of cephalosporin resistance (mainly via the spread of plasmids) in various Gram-negative bacteria such as *E. coli* [15]. 

An important finding in this field is that antimicrobial concentrations up to 230-fold lower than the minimal inhibitory concentration (MIC) are capable of inducing AMR in bacteria such as *E. coli* and *Salmonella enterica* spp. [23,24,25]. Whilst such experiments have not been conducted in *Neisseria* spp., concentrations of ciprofloxacin as low as 0.1 μg/L have been shown to select for resistance in other Gram-negative bacteria [23,26]. The minimum concentration of an antimicrobial that is able to select for antimicrobial resistance is defined as the minimum selective concentration (MSC) [23,26]. Quinolone concentrations in meat, water and environmental samples have been found to exceed this threshold in a number of locales. For example, studies have found the mean concentration of ciprofloxacin in samples of milk, eggs and edible fish in China to be 8.5 µg/L, 16.8 µg/kg and 331.7 µg/kg, respectively [27,28,29]. In addition, quinolones have been detected in tap water (median quinolone concentration 0.270 μg/L) [30] and bottled water [31] in China.

These findings provided the motivation for the two aims of this study: (i) assess the MSC of *N. gonorrhoeae* and (ii) assess at the country level if the prevalence of gonococcal ciprofloxacin resistance is associated with the intensity of quinolones used in food animal production. 

## 2. Materials and Methods

### 2.1. In Vitro Determination of Minimum Selection Concentrations (MSCs)

#### 2.1.1. Bacterial Strains and Growth Conditions

The WHO-P reference strain of *N. gonorrhoeae* (ciprofloxacin MIC 0.004 mg/L) was used in all experiments. The ciprofloxacin-resistant WHO-P used for the MSC_select_ experiment is an isogenic strain of WHO-P, except for the presence of a ciprofloxacin resistance-conferring mutation in the *gyrA* gene (S91F). 

All experiments were conducted using GC broth supplemented with IsoVitaleX™ (BD BBL™) (1%) for liquid or solid media. Strains were grown in an incubator at 36 °C in 6% CO_2_.

#### 2.1.2. MSC Determination

There are two important components of the MSC: (i) the minimum concentration of an antimicrobial at which one can induce de novo resistance (MSC*_denovo_*) and (ii) the lowest antimicrobial concentration that selects for a resistant—compared to a susceptible—strain (MSC_select_) [23,32]. The methodology to assess the MSC_select_ and the MSC*_denovo_* was based on the study by Gullberg et al. [23,26]. We defined ciprofloxacin resistance as the MIC ≥ 0.064 mg/L.

#### 2.1.3. MSC*_denovo_*

To assess if subinhibitory ciprofloxacin concentrations could select for de novo generated resistant mutants, susceptible WHO-P was exposed to a constant concentration of ciprofloxacin at 1, 1:10, 1:100 and 1:1000 of its ciprofloxacin MIC (0.004 mg/L) on GC agar plates. Identical control experiments were conducted with GC agar plates without ciprofloxacin. All strains were passaged to a new plate with the same conditions every 24 h for 7 days. The number of colonies of WHO-P with reduced susceptibility and resistance to ciprofloxacin was established by plating 600 µL of phosphate-buffered saline (PBS) solution containing the lawn of colonies (5.9–11.6 McFarland) from each experiment onto three GC agar plates as follows: (i) no ciprofloxacin, (ii) ciprofloxacin conc. of 0.016 mg/L (reduced susceptibility) and (iii) ciprofloxacin conc. of 0.06 mg/L. The number of colonies was counted after 24-h incubation at 36 °C. The lowest ciprofloxacin concentration that resulted in the growth of WHO-P, in the ciprofloxacin 0.016 mg/L or 0.06 mg/L plates and that contained at least one established fluoroquinolone resistance-associated mutation (S91F or D95N in *gyrA* [33]) was defined as the MSC*_denovo_*_._ Each experiment was conducted in quadruplicate. A single colony from each concentration was replicated on plates with the same concentration of ciprofloxacin they were isolated from to confirm their resistance, and additionally, MICs were determined via the E-test (bioMérieux, Marcy-l’Étoile, France).

#### 2.1.4. MSC_select_

Growth rates were measured in standard 15-mL vials in NGmorbidostat, which construction and optimization for the culture of *N. gonorrhoeae* has been described elsewhere [34,35]. In brief, NGmorbidostat vials are continuously stirred at a speed of 200 rotations per minute (rpm) via a magnetic stirrer. The optical density (OD) of each vial is measured once a minute via a system of infrared light-emitting diodes and photodetectors and recorded in an automated fashion (The Math Works, Inc., Natick, MA, USA. MATLAB, version R2015b).

The growth rates of the ciprofloxacin-susceptible (WHO-P) and -resistant (WHO-P S91F *gyrA* mutation; ciprofloxacin MIC 0.064 mg/L) *N. gonorrhoeae* at 36 °C in GC broth were assessed in vials containing the following ciprofloxacin concentrations: 0, 1, 1:10, 1:100 and 1:1000 times the ciprofloxacin MIC of WHO-P (0.004 mg/L). 

Each vial was inoculated with 100 µL of 4 McFarland WHO-P (one set with the susceptible and one with the resistant strain)*,* and the cultures were grown for 24 h. The relative growth rates at each concentration of ciprofloxacin were calculated as the observed growth rate of the strain divided by the growth rate of the same strain without ciprofloxacin. This experiment was conducted in triplicate.

Growthcurver was used in R (4.1.2) to summarize the growth characteristics of each vial [36]. Growthcurver fits a basic form of the logistic equation common in ecology and evolution to experimentally obtained growth curve data. It provides a number of summary measures to describe growth curves. We used two of these measures to calculate the MSC_select_:Growth rate. The intrinsic growth rate of the population, *r*, is the growth rate that would occur if there were no restrictions imposed on growth. Growthcurver uses the nonlinear least-squares Levenberg–Marquardt algorithm to determine *r.*Area under the curve. Growthcurver calculates the area under the logistic curve (AUC). This integrates information from the carrying capacity, growth rate and the population size at time 0.

The MSC_select_ was calculated as the point where the relative growth curves of the ciprofloxacin-resistant and -susceptible *Neisseria gonorrhoeae* crossed [26]. For the calculation of the MSC_select_, we calculated the average value obtained from the three individual experiments.

#### 2.1.5. Whole-Genome Sequencing

We confirmed the presence of mutations by whole-genome sequencing (WGS) of the samples: (i) A.MIC/100 (0.06 mg/L) and (ii) D.MIC/1000 (0.016 mg/L) (Appendix A). 

Genomic DNA was extracted using the DNeasy ^®^ Blood &Tissue Kit (Qiagen, Hilden, Germany) (t) and suspended in nuclease-free water (Sigma-Aldrich, Seelze, Germany). Paired-end 150-bp indexed reads using the Nextera XT DNA library prep kit were generated using Illumina technology according to the manufacturer’s instructions (Eurofins, Konstanz, Germany). The WGS data are available on GenBank (https://www.ncbi.nlm.nih.gov/sra/PRJNA798268). After the initial quality control by FastQC (https://github.com/s-andrews/FastQC) and trimming using trimmomatic [37], the processed Illumina reads were de novo assembled with Shovill (v1.0.4) (https://github.com/tseemann/shovill), which uses SPAdes (v3.14.0) (https://github.com/ablab/spades) using the following parameters: --rim --depth 150 --opts –isolate. The annotation was performed with Prokka (v1.14.6) (https://github.com/tseemann/prokka). The different single-nucleotide polymorphisms (SNPs) were determined using Snippy (v4.6.0) (https://github.com/tseemann/snippy). 

### 2.2. Ecological Association between Quinolone Use and Ciprofloxacin MICs

#### 2.2.1. Quinolone Use for Animal Food Production Data

We obtained the country-level consumption of quinolones for animal food production in the year 2013 from a systematic review on this topic performed by Broeckel et al. [38]. This study calculated the volume of antimicrobials (in tons) by class of antimicrobial in 38 countries in the year 2013. Four categories of animals were included: chicken, cattle, pigs and small ruminants (sheep and goats), which together account for the overwhelming majority of terrestrial animals raised for food [15,38]. 

We used this data to calculate the number of milligrams of quinolones used for animal food production/population correction unit (PCU) (a kilogram of animal product) in the year 2013. The data for the tonnage of food animals produced per country and year in the year 2013 was taken from the Food and Agriculture Organization estimates (http://www.fao.org/faostat/en/?#data/, accessed 18 February 2021).

#### 2.2.2. Quinolone Consumption in Humans

Data from IQVIA were used as a measure of the national antimicrobial drug consumption. IQVIA uses national sample surveys that are performed by pharmaceutical sales distribution channels to estimate the antimicrobial consumption from the volume of antibiotics sold in retail and hospital pharmacies [39]. The sales estimates from this sample are projected with the use of an algorithm developed by IQVIA to approximate the total volumes for sales and consumption [39]. Quinolone consumption (moxifloxacin, ciprofloxacin, emifloxacin, ofloxacin, levofloxacin, lomefloxacin, norfloxacin, enoxacin, gatifloxacin, trovafloxacin and sparfloxacin) estimates are reported as the number of standard doses (a dose is classified as a pill, capsule or ampoule) per 1000 population per year [2].

#### 2.2.3. *N. gonorrhoeae* Ciprofloxacin Resistance Data

The percent of isolates per country that were resistant to ciprofloxacin in the year 2014 (the year following the quinolone consumption variables) was taken from the WHO Global Gonococcal Antimicrobial Surveillance Programme (GASP; https://www.who.int/data/gho/data/indicators, accessed 18 February 2021). GASP is a collaborative global network of regional and subregional reference laboratories that monitors gonococcal AMR in participating countries. The full GASP methodology, including suggested sampling strategy, laboratory techniques, external quality assurance and internal quality control mechanisms published elsewhere [40]. GASP uses a minimum inhibitory concentration (MIC) breakpoint of 1 μg/mL to define resistance to ciprofloxacin, which was, therefore, the definition of ciprofloxacin resistance we used [40]. In the case of 4 countries, data were not available for the year 2014. We used the data for the first subsequent year with available data—2016 (Czechia, Luxembourg) and 2017 (Finland, Switzerland). 

#### 2.2.4. Statistical Analysis

Spearman’s correlation was used to assess the association between the prevalence of ciprofloxacin resistance in *N. gonorrhoeae* and the two independent variables—quinolone use for animals and quinolone consumption by humans for the purpose of treating infections. Linear regression was used to assess the country-level association between the percent of *N. gonorrhoeae* isolates with ciprofloxacin resistance and the two independent variables in 3 models. We started by assessing the association between ciprofloxacin resistance and quinolone consumption in humans (Model 1). We then assessed the association between ciprofloxacin resistance and quinolone use in animals (Model 2). Finally, in Model 3, we evaluated the effect of both independent variables on ciprofloxacin resistance. Stata 16.0 was used for all analyses. A *p*-value of <0.05 was considered statistically significant. 

## 3. Results

### 3.1. Minimum Selection Concentration

#### 3.1.1. MSC*_denovo_*

Ciprofloxacin concentrations as low as 0.00004 mg/L (1/100th the MIC) were able to induce resistance to ciprofloxacin in a fully susceptible strain of WHO-P (Figure 1; Appendix A). Out of the four experiments, three showed growth at this concentration. Whole-genome sequencing in one of the ciprofloxacin-resistant colonies revealed that it acquired an E137K substitution in *porB*. This mutation has not been linked to fluoroquinolone resistance in *N. gonorrhoeae* previously. 

Ciprofloxacin concentrations of 1/1000th the MIC induced a four-fold increase in MIC (to 0.016 mg/L; Figure 1; Appendix A), but this occurred in only one colony in one of the four experiments. Whole-genome sequencing of this colony revealed that it had acquired the *gyrA* D95N mutation.

Higher concentrations of ciprofloxacin resulted in a higher number of ciprofloxacin resistant colonies (Figure 1; Appendix A). 

#### 3.1.2. MSC_select_

The mean MSC_select_ obtained by the AUC ratio method (mean 0.007 µg/L, range 0.001–0.019 µg/L) was considerably lower than that obtained by the growth rate method (mean 0.413 µg/L, range 0.023–0.98 µg/L; Figure 2; Appendix A).

#### 3.1.3. Ecological Association between Quinolone Consumption and Ciprofloxacin Resistance

No animal quinolone consumption data were available for two countries (Iran and Nepal). There was considerable variation in the intensity of quinolone use for food animals in the 36 countries with data (median 1.9 mg quinolones/PCU; IQR 0.7–6.6 mg/PCU; Appendix A). Quinolone exposure in food animals in China was higher than all other countries (261.2 mg/PCU). 

The prevalence of ciprofloxacin resistance in *N. gonorrhoeae* was significantly positively associated with quinolone use for food animals (ρ = 0.47; *p* = 0.004; N = 34) but not quinolone consumption in humans (ρ = 0.31; *p* = 0.097; N = 30). 

The regression model that combined quinolone use in food animals and humans was a better predictor of gonococcal ciprofloxacin resistance (R^2^ = 0.30; Model 3) than the models that only included either quinolone use in animals or humans (R^2^ = 0.14 (Model 2) and R^2^ = 0.07 (Model 1), respectively; Table 1).

## 4. Discussion

We found that the gonococcal MSC for ciprofloxacin was lower than that for any other bacteria on record [23,26]. In particular, the MSC*_denovo_* (0.004 µg/L) was considerably lower than quinolone concentrations found in foodstuffs such as meat products, milk and water in China and elsewhere [27,28,29,30,31]. The consumption of foods with high quinolone concentrations has been found to be associated with high urinary and fecal concentrations of quinolones in humans [41,42,43,44]. For example, a study from South Korea found that high urinary excretions of enrofloxacin and ciprofloxacin in the general population were strongly associated with the consumption of beef, chicken and dairy products [41]. Similarly, a large study of the general population in three regions of China found ciprofloxacin, enrofloxacin and ofloxacin in the feces of 67%, 30% and 57% of individuals [44]. The authors attributed the high median concentration of quinolones (median 20 μg/kg) in large part to the ingestion of veterinary antimicrobials in food [45]. Reducing the consumption of these foodstuffs has also been found to result in a reduction of the urinary quinolone concentrations [45]. The problem of high antimicrobial concentrations in human food is not limited to Asian countries. In European countries, food animals consume a greater mass of antimicrobials per kilogram per year than humans do [46]. Furthermore, studies from Europe have shown that AMC in animals is independently associated with AMR in bacterial species colonizing and infecting humans [46]. The concentration of quinolones varies considerably by locale and meat type. We were unable to find a systematic review on the topic. In addition to the studies noted above, we found studies where the mean concentration of ciprofloxacin in chicken meat was 6.3 μg/kg (Lebanon) and 31 μg/kg (Turkey) and, in goat’s milk, was 100 mcg/L (Spain) [1,2,3]. 

These findings generate the hypothesis that quinolone concentrations in meat may play a role in the genesis of quinolone resistance in bacteria such as *N. gonorrhoeae* [47,48,49]. We found supportive evidence at the country level—quinolone consumption for animal husbandry was positively associated with the prevalence of ciprofloxacin resistance in *N. gonorrhoeae*. Furthermore, the combination of quinolone consumption in humans and food animals provided the best prediction of ciprofloxacin resistance. These findings could explain the higher prevalence of quinolone resistance in *N. gonorrhoeae*, *N. meningitidis* and commensal *Neisseria* in countries such as China than would be predicted based on the human consumption of quinolones [2,6,40,50]. 

Our estimates of the MSC_select_ derived from the two methods differed considerably. The estimate of the MSC_select_ derived from the AUC ratio method was, however, similar to the MSC*_denovo_*. A number of factors have been shown experimentally to affect the MSC. For example, MSCs have been shown to increase by 13- to 43-fold in complex environments, such as those in the human body where bacteria compete with one another and interact with the host’s defenses [24,51,52]. The use of *gfp*-labeling of the resistant and susceptible strains in direct competitive growth assays has also been shown to provide a more accurate way to ascertain the MSC_select_ [26]. 

Furthermore, in a real-world setting, human microbiota would be exposed to low-dose antimicrobials intermittently (such as during and after eating) rather than continuously, such as in our experiments. There are also large differences in antimicrobial penetration to different anatomical sites, such as the oropharynx, which we have not considered [53]. We also do not know if the effect of quinolones in food on *Neisseria* would act via direct contact with *Neisseria* in the mouth during mastication, in the colon during drug elimination or in the genital tract following absorption and distribution. We have also not considered the indirect commensal pathway through which quinolones could select for AMR. Quinolones could select for resistance in commensals that could then be transferred to the pathogenic *Neisseria* via transformation [6,54]. These considerations imply that our estimates of the gonococcal MSC should be viewed as tentative. It is possible and even likely that the lowest concentration that can select for quinolone resistance in *Neisseria* species is contingent on a large number of cofactors such as microbiome community state types and individual human pharmacogenomic variations [24,51,52,55].

For these reasons, we consider the calculation of an exact gonococcal ciprofloxacin MSC less important than establishing that ciprofloxacin concentrations considerably lower than the MIC can select for resistance. Our results suggest the need for experiments to ascertain if the concentrations of quinolones detected in contemporary foodstuffs can induce AMR in *N. gonorrhoeae* and other bacteria. Whilst it may be difficult to perform human challenge studies with *N. gonorrhoeae*, studies with commensal *Neisseria* may be possible. A further option would be to assess if food spiked with low concentrations of quinolones could induce AMR in *Neisseria musculi*, a colonic and oropharyngeal commensal of the common mouse, *Mus musculus* [56,57]. These studies may be of considerable use in determining what safe maximum residue limits (MRLs) of quinolones are in foodstuffs. The current European Commission and WHO/FAO guidelines for establishing MRLs evaluate the effect of antimicrobial residues on the toxicity towards a range of bacteria [58,59]. However, they do not evaluate the effect of these residues on the genesis of AMR [52,58,60,61]. 

There are also a number of limitations pertaining to the ecological study. These include the relatively small number of countries with available data, the lack of longitudinal data on quinolone consumption in animals and the absence of data on quinolone use for aquaculture. The epidemiology of resistance is complex, and factors other than the mass of quinolones consumed may influence the level of quinolone resistance. These include the consumption of other classes of antimicrobials, inadequate sanitation, travel by humans and trade of live animals and meat [15]. The movement of humans, animals and meat may be particularly important factors, since these can result in the dissemination of bacteria and genes conferring AMR [15].

We found that the E137K substitution in PorB emerged under low-dose ciprofloxacin selection pressure. This was the only plausible mutation that may explain the elevated ciprofloxacin MIC of this isolate. We could not find any evidence in the literature that this mutation has been linked to ciprofloxacin susceptibility. To explore the possibility that this mutation may result in reduced ciprofloxacin susceptibility, we searched for this mutation in a collection of 17,881 *N. gonorrhoeae* isolates publicly available at Pathogenwatch (https://pathogen.watch/) and PubMLST (https://pubmlst.org/), accessed on 2 April 2020. We found that E137K substitution was present in 311 isolates (1.7%). Out of the 311 isolates, ciprofloxacin MIC was not available for 286 isolates (91.9%), whereas 22 (7.07%) and 3 (1%) isolates were resistant and susceptible, respectively. In comparison, 15,923 (89%) isolates had glutamine at position 137. Of these, 11,141 (69.9%) had no ciprofloxacin MIC available, and 3014 (18.9%) and 1768 (11.1%) isolates were resistant and susceptible, respectively. The G120K and A121D/N mutations in PorB have previously been linked to penicillin, cephalosporin and tetracycline resistance [33,62]. These positions are situated in loop 3 of the protein, which modulates the permeability of the porin to hydrophilic antimicrobials [62]. Whilst speculative, mutations such as E137K, which also occur in loop 3, may modulate permeability in a similar fashion.

Our results are thus best considered hypothesis-generating. Further in vivo experiments along the lines outlined above will be required to assess if quinolones in food residues play a role in the genesis of quinolone resistance in *N. gonorrhoeae* and related bacteria.

## 5. Conclusions

This study had two aims. Firstly, we undertook to ascertain the MSC of *N. gonorrhoeae*. Our experiments revealed that ciprofloxacin concentrations 1000 times lower than the MIC could induce a four-fold increase in MIC. Whole-genome sequencing revealed that this increase in MIC was associated with the *gyrA* D95N mutation, which is a well-established cause of quinolone resistance. In addition, we found a novel E137K mutation in *porB,* which may be associated with reduced susceptibility to ciprofloxacin. Secondly, we aimed to assess at the country level if the prevalence of gonococcal ciprofloxacin resistance is associated with the intensity of quinolones used in food animal production. We found that the prevalence of gonococcal resistance to ciprofloxacin was positively associated with the intensity of quinolone use in food-producing animals.

Further individual level research is required to assess if low doses of ciprofloxacin from ingested foodstuffs are able to select for ciprofloxacin resistance in *N. gonorrhoeae* and other species.

## Figures and Tables

**Figure 1 antibiotics-11-01430-f001:**
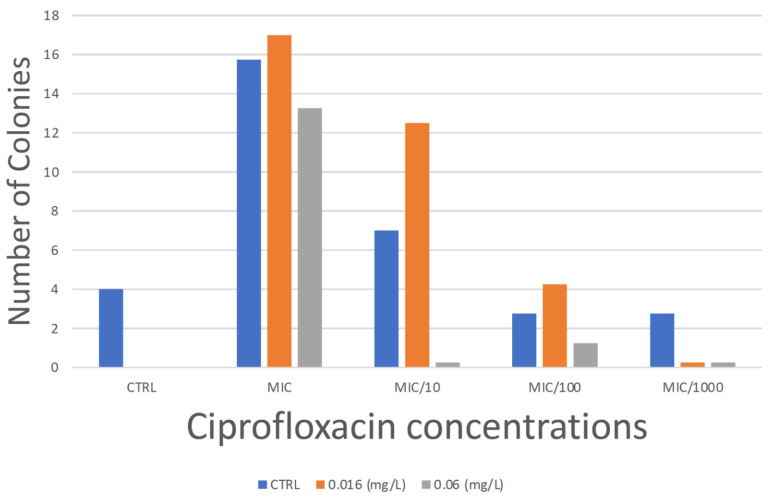
Growth distribution of colonies. Mean number of colonies of *N. gonorrhoeae* grown at 7 days in different ciprofloxacin concentrations ranging from 1xMIC to 1/1000th MIC of WHO-P, stratified according to their final growth in plates with no ciprofloxacin, 0.016 mg/L and 0.06 mg/L. All the experiments were conducted in quadruplicate.

**Figure 2 antibiotics-11-01430-f002:**
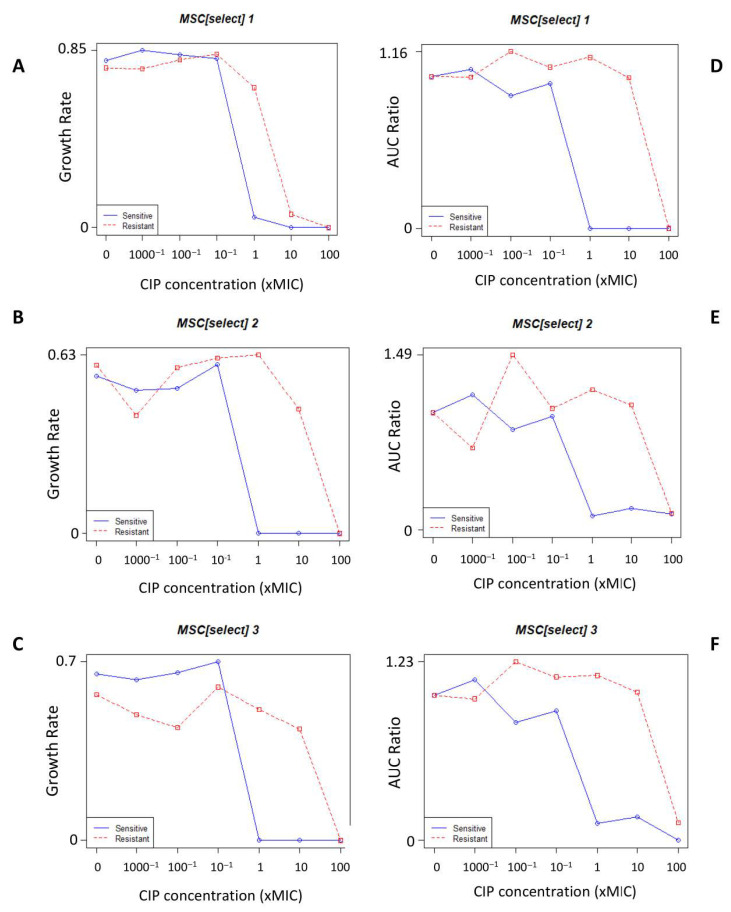
Growth rates as a function of ciprofloxacin (CIP) concentrations. The relative exponential growth rates (**A**–**C**) and AUC ratio (**D**–**F**) of *N. gonorrhoeae* WHO-P with (red) and without (blue) the *gyrA* S91F mutation conferring resistance to ciprofloxacin. The MSC_select_ is determined as the point where the blue and red lines intersect.

**Table 1 antibiotics-11-01430-t001:** Linear regression models testing the country-level association between quinolone consumption in food animals/humans and the prevalence of ciprofloxacin resistance expressed as a percentage (coefficients (95% confidence intervals)).

	Model 1	Model 2	Model 3
Quinolones Humans	0.02 (−0.007–0.043)	-	0.2 (0.001–0.046) *
Quinolones Food animals	-	0.2 (0.02–0.38) *	0.2 (0.07–0.40) **
N	30	34	30
R^2^	0.07	0.14	0.30

* *p*-value < 0.05 and ** *p*-value < 0.01.

## Data Availability

GenBank (https://www.ncbi.nlm.nih.gov/sra/PRJNA798268).

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
