# Peer review of "Ciprofloxacin Concentrations 1/1000th the MIC Can Select for Antimicrobial Resistance in N. gonorrhoeae—Important Implications for Maximum Residue Limits in Food"

_antibiotics, 2022, doi:10.3390/antibiotics11101430_

Round 1
Reviewer 1 Report
The manuscript presents a very relevant theme, which is the impact of the use of antimicrobials in animal production in the selection of resistance to an agent of importance in human medicine. I believe that the scientific approach given to the subject is quite pertinent and adequate.
Figures in graph form are difficult to visualize and could be improved.
The antimicrobial consumption analysis data in different countries is a little old (2013), could be updated.
Reviewer 2 Report
Please see the attached document - with my comments.

Reviewer 3 Report
Unfortunatly, based on the conclusion of the manuscript, the research has not any significant output. The authors mentioned about 2 objectives in the introduction, but in the conclusion, there is no justification about their hypothesis.
